# Hepatocellular Carcinoma and the Multifaceted Relationship with Its Microenvironment: Attacking the Hepatocellular Carcinoma Defensive Fortress

**DOI:** 10.3390/cancers16101837

**Published:** 2024-05-11

**Authors:** Linda Galasso, Lucia Cerrito, Valeria Maccauro, Fabrizio Termite, Maria Elena Ainora, Antonio Gasbarrini, Maria Assunta Zocco

**Affiliations:** 1Department of Internal Medicine and Gastroenterology, Fondazione Policlinico Universitario Agostino Gemelli IRCCS, Catholic University of Rome, 00168 Rome, Italylucia.cerrito@policlinicogemelli.it (L.C.); valeriamaccauro@gmail.com (V.M.); antonio.gasbarrini@policlinicogemelli.it (A.G.); 2CEMAD Digestive Disease Center, Fondazione Policlinico Universitario Agostino Gemelli IRCCS, Catholic University of Rome, 00168 Rome, Italy

**Keywords:** hepatocellular carcinoma, microenvironment, therapy

## Abstract

**Simple Summary:**

The microenvironment of hepatocellular carcinoma is a really complex milieu, containing a wide variety of cells with specific tasks, belonging to both the mesenchimal and immunitary systems, the extracellular matrix, growth factors, proinflammatory cytokines, and translocated bacterial products. This intricate environment represents a source of potential targets in order to establish a wide and multitask therapeutic strategy for hepatocellular carcinoma.

**Abstract:**

Hepatocellular carcinoma is a malignant tumor that originates from hepatocytes in an inflammatory substrate due to different degrees of liver fibrosis up to cirrhosis. In recent years, there has been growing interest in the role played by the complex interrelationship between hepatocellular carcinoma and its microenvironment, capable of influencing tumourigenesis, neoplastic growth, and its progression or even inhibition. The microenvironment is made up of an intricate network of mesenchymal cells, immune system cells, extracellular matrix, and growth factors, as well as proinflammatory cytokines and translocated bacterial products coming from the intestinal microenvironment via the enterohepatic circulation. The aim of this paper is to review the role of the HCC microenvironment and describe the possible implications in the choice of the most appropriate therapeutic scheme in the prediction of tumor response or resistance to currently applied treatments and in the possible development of future therapeutic perspectives, in order to circumvent resistance and break down the tumor’s defensive fort.

## 1. Introduction

Primary liver cancer is ranked as the sixth most diagnosed cancer globally, while also standing as the third and second leading cause of cancer-related death and premature cancer-related death worldwide, respectively. Projections suggest a 55% increase in its incidence over the next two decades, highlighting its significant impact on global health [1]. Among primary liver cancers, hepatocellular carcinoma (HCC) emerges as the dominant subtype, accounting for roughly 80% of cases [2,3].

The primary risk factor contributing to the onset of HCC is the presence of fibrotic or cirrhotic conditions, often stemming from chronic inflammatory causes such as infection with hepatitis B virus (HBV) or hepatitis C virus (HCV), alcohol-related liver disease (ALD), or metabolic dysfunction-associated steatotic liver disease (MASLD) [4]. Less common risk factors include hereditary hemochromatosis, primary sclerosing cholangitis, primary biliary cholangitis, autoimmune hepatitis, and other chronic hepatitis variants [5]. Moreover, epidemiological investigations have illuminated associations between environmental exposures, such as aflatoxin, smoking, and air pollution, and an elevated susceptibility to HCC [6]. Therefore, the epidemiological profile of HCC reflects the distribution of these underlying causes: changes in the global incidence patterns of HCC over time correspond with the initiation of national vaccination campaigns against HBV; the advent of direct-acting antiviral agents (DAAs) for managing HCV infection; and fluctuations in alcohol consumption and obesity rates, impacting the prevalence of ALD and MASLD, respectively. According to findings from the Global Burden of Disease study, approximately 41.0% of HCC cases diagnosed in 2019 were linked to HBV infection, 28.5% to HCV infection, 18.4% to ALD, 6.8% to MASLD, and 5.3% to other less frequent risk factors [7].

Despite notable advancements in therapy, HCC presents one of the most unfavorable prognoses, with a five-year survival rate ranging from 15% to 38% [8,9]. This is largely due to delayed diagnosis, resistance to chemotherapy, frequent recurrence, and metastasis.

Treatment modalities such as liver transplantation, surgical resection, radiofrequency ablation, transarterial chemoembolization, and radiotherapy are effective for liver-located HCC. Conversely, systemic therapy targeting the tumor microenvironment (TME) is an option for cases of unresectable HCC. The TME is a vital ecosystem that orchestrates key events from cancer initiation to distant metastasis. The microenvironment comprises a complex interplay of mesenchymal cells, immune cells, extracellular matrix components, and growth factors, as well as pro-inflammatory cytokines and translocated bacterial products originating from the intestinal microenvironment through the enterohepatic circulation [10,11,12].

This paper aims to depict the role of the HCC microenvironment, a deeply complex cellular and molecular system, and its potential implications for selecting the most suitable treatment strategies (predicting tumor response or resistance), and envisioning future therapeutic approaches to circumvent resistance and break down the tumor’s defensive fort.

## 2. Systemic Therapy in Advanced HCC

### 2.1. Multikinase Inhibitors

In 2007, findings from the phase III SHARP trial described the survival advantages of sorafenib compared to placebo (median overall survival 10.7 months versus 7.9 months) [13], establishing it as the first drug approved for the treatment of unresectable HCC and the only systemic therapeutic option available until 2020. The effectiveness of sorafenib likely stems from its ability to target both cancer cells and cells within the TME, exhibiting inhibitory activity against approximately 40 kinases, predominantly angiogenic receptor tyrosine kinases (RTKs) like vascular endothelial growth factor receptors (VEGFRs) and platelet-derived growth factor receptor-beta (PDGFRβ), as well as drivers of cell proliferation such as RAF1, BRAF, and KIT [14].

A decade later, a new targeted therapy, lenvatinib, emerged as a first-line treatment option for individuals with unresectable HCC [15]. This oral multikinase inhibitor acts on VEGFR1-3 and fibroblast growth factor receptors (FGFR1-4), PDGFR, KIT, and RET [16].

The multikinase inhibitors family also encompasses regorafenib, ramucirumab and cabozatinib, drugs that are currently approved as second-line treatments, after progression in patients treated with sorafenib (cabozantinib may also be prescribed as a third-line treatment) [17]. Regorafenib shares structural similarities with sorafenib; however, their inhibitory profiles exhibit slight differences: regorafenib demonstrates enhanced potency against VEGFR kinases and boasts a broader spectrum of activity, including targets such as angiopoietin 1 receptor (TIE2), KIT, and RET [18]. Cabozantinib is a small-molecule drug that demonstrates potent inhibition of MET and AXL, alongside its activity against VEGFRs [19], while ramucirumab is an antagonistic anti-VEGFR2 monoclonal antibody [20].

### 2.2. Immune-Checkpoint Inhibitors

The influence of therapies aimed at immune checkpoints on oncological strategies is currently relevant. Agents targeting cytotoxic T-lymphocyte antigen 4 (CTLA-4), programmed cell death protein 1 (PD-1), or programmed cell death protein ligand 1 (PD-L1) have transformed the approach to the management of numerous types of tumors, including HCC: in the landscape of advanced HCC systemic treatments, the combination of atezolizumab and bevacizumab (Atezo-Bev) actually stands as the first-line therapy, demonstrating superior survival outcomes compared to sorafenib [13,17,21].

Atezolizumab is a monoclonal antibody against PD-L1. PD-1, a receptor present on T cells, plays a critical role in providing inhibitory signals primarily during the effector stage of T-cell responses. Within the context of cancer development, PD-1 expressed on T cells interacts with its two identified ligands, PD-L1 and PD-L2, within the TME, leading to the suppression of anti-tumoral immune responses [22]. On the other hand, bevacizumab falls outside the family of immune checkpoint inhibitors as it is a monoclonal antibody that binds to vascular endothelial growth factor (VEGF), inhibiting its interaction with receptors and thus neutralizing its biological activity: promoting angiogenesis, regulating vascular permeability, hindering dendritic cell maturation, promoting infiltration of immune suppressive cells, and upregulating expression of immune checkpoint molecules [23].

In more recent studies, the combination therapy of durvalumab and tremelimumab demonstrated superior activity compared to sorafenib, exhibiting a higher response rate and improved overall survival (median: 16.4 months versus 13.8 months), but with no impact on progression-free survival [24]. Durvalumab shares the same molecular target as atezolizumab (anti-PD-L1), while tremelimumab is a monoclonal antibody against CTLA-4. CTLA-4, constitutively expressed in regulatory T cells and upregulated in cytotoxic T cells following T-cell priming, acts as a dominant negative signaling molecule. Monoclonal antibodies targeting CTLA-4 effectively inhibit this negative feedback mechanism, resulting in significant and lasting responses in cancer patients [25].

## 3. Results

This section, possibly subdivided by relevant headings, aims to present a clear and concise summary of the experimental results, their interpretation, and the conclusions drawn from the experiments.

### 3.1. Hepatocellular Carcinoma and the Tumor Microenvironment

The microenvironment of HCC is intricate and ever-changing, marked by the interplay among tumor cells and diverse elements of adjacent tissue (Figure 1). This milieu significantly influences tumor onset, advancement, and treatment outcomes. It encompasses a heterogeneous mix of stromal cells, such as cancer-associated fibroblasts (CAFs), endothelial cells, and immune cells.

CAFs, a prominent constituent of the tumor microenvironment (TME) in HCC, exert significant effects on tumor progression, invasion, and metastasis through paracrine signaling and extracellular matrix (ECM) remodeling. Their activation is orchestrated by various stimuli, such as transforming growth factor-β (TGF-β) and platelet-derived growth factor (PDGF) [26].

Endothelial cells form crucial blood vessels that sustain tumor viability by providing nutrients and oxygen, while also facilitating metastatic dissemination [27].

The immune dynamics within HCC are characterized by a complex interplay involving various cellular components such as tumor-infiltrating lymphocytes (TILs), myeloid-derived suppressor cells (MDSCs), regulatory T cells (Tregs), and tumor-associated macrophages (TAMs). TILs, particularly cytotoxic CD8+ T cells, play a crucial role in monitoring and eliminating tumors [28].

However, the presence of immunosuppressive elements like MDSCs, Tregs, and TAMs creates an immune-suppressive environment leading to tumor immune evasion and disease progression [11].

Additionally, the HCC microenvironment contains cancer stem cells (CSCs), a subset of tumor cells possessing self-renewal and differentiation abilities. These CSCs significantly contribute to tumor heterogeneity, therapy resistance, and disease recurrence in HCC. Crosstalk between cancer cells and stromal components within the TME contributes to aberrant Wnt signaling promoting tumor proliferation, invasion, and metastasis [29,30,31].

Within the hepatocellular carcinoma (HCC) microenvironment, the extracellular matrix (ECM) assumes a pivotal role, providing structural integrity to the tumor and orchestrating cellular behavior through signaling molecules such as growth factors and cytokines. Alterations in ECM composition and stiffness can profoundly impact tumor dynamics, fostering growth, invasion, and metastasis [32,33].

Notably, rapid tumor proliferation often leads to insufficient vascularization and subsequent hypoxia, triggering the secretion of angiogenic factors like vascular endothelial growth factor (VEGF) and hypoxia-inducible factor 1-alpha (HIF-1α). This cascade culminates in the formation of new blood vessels, vital for tumor sustenance and metastatic spread [34]. Extracellular vesicles (EVs), comprising exosomes and microvesicles, emerge as significant players in the HCC microenvironment. Originating from both tumor and stromal cells, these vesicles carry a cargo of proteins, nucleic acids, and lipids that facilitate intercellular communication, modulating diverse aspects of tumor progression, encompassing immune regulation, angiogenesis, and metastasis [35].

Also, microbes play a significant role in shaping the hepatocellular carcinoma (HCC) microenvironment, particularly through the intricate interplay of the gut microbiota and the liver, known as the gut–liver axis. Dysbiosis disrupts this axis, leading to heightened intestinal permeability and bacterial translocation. Consequently, dysbiosis-induced shifts in metabolite production can modulate inflammatory signaling pathways, cellular proliferation, and apoptosis, all pivotal in HCC pathogenesis [12,36]. Notably, microbial products and metabolites originating from the gut can exert direct influences on liver function and hepatocyte behavior. For instance, lipopolysaccharide (LPS) activation of hepatic stellate cells triggers the production of pro-inflammatory cytokines, initiating liver fibrosis, a precursor to HCC. Studies have pinpointed specific alterations in gut microbiota composition associated with HCC, characterized by diminished microbial diversity, proliferation of pathogenic bacteria such as Escherichia coli and Fusobacterium nucleatum, and reduced levels of beneficial bacteria like Bacteroides and Lactobacillus [12].

Understanding the intricate interplay within the HCC microenvironment offers potential avenues for therapeutic interventions targeting both tumor and stromal components, as well as microbial influences, to improve patient outcomes.

### 3.2. Monocytes

The monocytes are attracted in the HCC microenvironment through specific stomal chemokines (CCL2 (chemokine (C-C motif) ligand 2), CCL15 (chemokine (C-C motif) ligand 15)) [37,38,39].

They can be differentiated in three subgroups (CCR1 + CD14+, CD14+, myeloid-derived suppressor cells—MDSCs), all promoting an immunosuppressive milieu that expresses specific cytokines (CXCL2, CXCL8, IL-10) [40] and immune-checkpoint inhibitors (PD-L1/2, TIM3), with an inducing effect over regulatory T cells (Tregs) and an inhibitory effect over natural killer (NK) cells [41].

Tumor cells prevent the differentiation of monocytes to macrophages, remaining in an immature state that contributes to create an immunotolerant environment: through the production of TGF-β, they can suppress the proliferation of T lymphocytes and induce Tregs, as well as expanding neoangiogenesis via the production of VEGF and MMP9 [39,42]. Another crucial mechanism is represented by the interaction with the neutrophils, mediated by CXCL2/CXCL8, enhancing neoplastic invasiveness through the production of the pro-metastatic factor oncostatin M [40].

Li et al. state that the inhibition of CCL15 pathway could be useful to block tumor development through the prevention of monocyte recruitment and of Treg differentiation [38]. In their study, Chang et al. noted that in orthotopic liver tumors undergoing sorafenib treatment, the infiltration of LY6G+ MDSCs was associated with a marked rise in IL-10 and TGF-β expression in CD4+ T cells, alongside a concurrent inhibition of the cytotoxic activity of CD8 T cells. Furthermore, they observed that IL-6 provided protective effects for LY6G+ MDSCs against sorafenib-induced cell death in vitro. Combining either anti-LY6G+ antibody or anti-IL-6 antibody with sorafenib resulted in a notable decrease in the presence of LY6G+ MDSCs within orthotopic liver tumors. This combined approach also led to increased T-cell proliferation and demonstrated a synergistic enhancement in the therapeutic effectiveness of sorafenib [39,43].

### 3.3. Macrophages

In HCC, macrophages are one of the main elements of TME and are responsible for the enhancement of cancer stem cells, angiogenesis, metastatization process, immune suppression, and resistance to anti-neoplastic drugs [44,45,46]: in fact, several studies observed that increased TAM infiltration in the tumor or at the border of the neoplastic mass is associated to poor prognosis after HCC resection [47] (Table 1).

Tumor-associated macrophages produce IGF-1 and hepatocyte growth factor (HGF), determining the proliferation of neoplastic cells and the recruitment of monocytes and the proliferation of neoplastic cells [48]. In particular, the HCC microenvironment is characterized by macrophages that are “programmed” by the tumor in an immunosuppressive way, due to the tumoral production of osteopontin and CCL2 [49]. The macrophages are also able to express CSF-1, leading to the enhancement of the tumoral programmed death-ligand 1 (PD-L1), thus increasing the immunosuppressive environment advantageous for tumoral proliferation [39,49].

Moreover, such a particular immunological milieu is favorable to the expression of M2 cells, which produce cytokines such as vascular endothelial growth factor (VEGF), IL-6, and L-10 that determine the inhibition of macrophages, NK cells, and T cells, as well as favoring the differentiation of particular regulatory T cells (Tregs) that are involved in the mechanisms of resistance to neoplastic treatments [50,51].

On the therapeutic side, besides the guideline-approved systemic therapies, which mainly regard mechanisms interfering with the immune-checkpoint and involving the lymphocytes, several strategies have been attempted to remodulate TAM activity.

There are two potential strategies involving macrophages in anti-cancer treatment: the first is the inhibition of their recruitment and differentiation, and the second is targeting chronic inflammation by inhibiting their activity. These therapeutic solutions gave important effects, particularly if combined with other therapeutic agents (antiangiogenetic, cytoreductive, or immunotherapic factors), while the researchers investigated the possibility to neutralize the mediators involved in macrophage recruitment such as elements of the complement cascade, chemokines, vascular endothelial growth factor (VEGF), c-MET, HGF, VEGF, colony-stimulating factor-1 (CSF-1), CCL-2, and the IL-6 pathway [52,53].

A study by Yang et al. on mice HCC models proposed 17-βestradiol as an agent that is able to suppress macrophage alternative activation, directing macrophage differentiation from M2-type to M1-type, consequently inhibiting HCC progression and decreasing its growth and metastatic spread through the enhancement of anti-neoplastic immune response and local vascular normalization [54]: its mechanism of action consists in preventing the interaction between estrogen-receptor ERβ and ATP-ase-coupling factor 2, with the subsequent inhibition of the signaling pathway JAK1-STAT6. Estrogens may represent a complementary therapy for HCC, especially in male subjects, in combination with therapeutic molecules specifically targeting cancer cells [39,54].

Colony-stimulating factor-1 (CSF-1) is one the main chemoattractants in the neoplastic environment, determining an attraction of macrophages to TME and the shift of their differentiation to a pro-tumorigenic type [55]: Ao et al. documented in both allograft and xenograft mouse hepatoma models that PLX3397, a competitive inhibitor of the receptor of colony-stimulating factor-1 (CSF-1R), is able to delay tumor mass increase through macrophage transition from M2-type to M1-type, thus prolonging the survival of tumor-affected mice. CSF-1R is involved in the regulation of the differentiation of macrophage function and in their infiltration in HCC. The study by Ao et al. was not able to demonstrate an effect of PLX3397 on macrophagic intratumoral infiltration in HCC tissue, also showing an increase in infiltration by CD8+ T cells and decrease in CD4+ T cells. Furthermore, CSF-2 produced by the neoplastic cells was able to prevent a PLX3397-mediated decrease in tumor-associated macrophages. The effects of the blockade of CSF-1R in human HCC is still uncertain, and further clinical trials are needed to prove its role in a potential combination therapy [56].

Fu et al. analyzed 26 patients with HCC who underwent chemoembolization with oxaliplatin within three months before radical resection: at the histological examination, the authors observed that the density of TAM in HCC was related to the efficacy of intraarterial treatment, because TAM impaired the death of HCC cells that should have been induced by oxaliplatin, thus demonstrating the involvement of TAMs in the establishment of drug resistance mechanisms. The authors suggest that using TAMs as a therapeutic target could increase the response to TACE with oxaliplatin [57].

Another possible therapeutic target is represented by the Wnt/b-catenin signaling pathway, because it promotes the polarization of M2 macrophages in HCC TME. For this reason, blocking WNT/b-catenin pathway in TAMs may rescue immune evasion of HCC [39,58].

**Table 1 cancers-16-01837-t001:** The main studies exploring the role of macrophages in tumor microenvironment for the treatment of hepatocellular carcinoma.

Author	Subjects	Agent	Target	Effects
Yang et al. [54]	Mice	17-βestradiol	Macrophages	- attenuation of HCC progression - decrease in HCC growth and metastatic spread - enhancement of anti-neoplastic immune response - local vascular normalization
Ao et al. [56]	Hepatoma models mice	PLX3397	Colony-stimulating factor-1 (CSF-1R)	- delay of tumor mass increase (macrophage transition from M2-type to M1-type) - prolong survival - differentiation of macrophages and their infiltration in HCC
Fu et al. [57]	Humans (26 patients)	Chemoembolization with oxaliplatin	Tumor-associated macrophages	- influence of HCC cells death (drug resistance mechanisms)
Yang et al. [58]	M2-like phenotype of TAMs in coculture with Hepa1-6 HCC cells	Wnt ligands	WNT/β-catenin pathway	- M2 macrophages polarization in HCC TME - Rescue of HCC immune evasion

### 3.4. Dendritic Cells

Dendritic cells (DCs) represent the connection between the cells of innate and adaptive immunity, with an essential role in the activation of anti-tumor adaptive immune response and in the initiation of T cells against neoplastic antigen that are involved in the progression of HCC. They are one of the main elements in the antigen-presenting process: they use MHC class II molecules to present the antigen to CD4+ T helper lymphocytes and MHC class I to present the antigen to CD8+ T lymphocytes. The contemporary interplay with costimulatory elements belonging to the TNF category and immunoglobulin family (CD80 and CD86, which bind to CD28 on T cells) is responsible for cytokine release (IFN, IL-12) and T lymphocyte activation and differentiation [59].

A specific subset of DCs, plasmacytoid DCs (pDCs), are associated with autoimmunity and IFN-I and IL-12 secretion, which facilitates the polarization of CD4+ T cells into Th1 helper T cells and the consequent accumulation of CTL CD8+ T cells involved in anti-tumor response [60].

HCC cells have the capability to prompt immature differentiation of DCs through the secretion of immunosuppressive factors like IL-10, IL-6, TGF-β, and VEGF. These immature DCs foster tumor tolerance by facilitating the development of CD8+ Treg cells and impeding the function of other effector T cells through a reduction in IL-12 secretion [61,62].

CD14 + CTLA-4+ tolerogenic DCs are a regulatory DC subtype that can suppress the anti-tumor immune response of CTLs via IL-10 and indoleamine-2,3-dioxygenas [63].

Targeting DCs represents a promising new therapy to foster anti-cancer immune response: in particular, DCs transfected with total HCC mRNA stimulated an antigen-specific cytotoxic T cell response capable of recognizing and killing autologous tumor cells in vitro [64].

Ormandy et al. emphasized that the dysfunction of DCs caused by IL-12 deficiency could lead to diminished activation of naïve T cells, proposing that directed IL-12 therapy might bolster tumor-specific immune reactions in individuals with HCC [65]. The overexpression of IL-12 induced via adenovirus vector has been shown to successfully prime DCs. It is crucial for effective tumor regression that activated IL-12-DCs are injected directly into the tumor site rather than systemically. Enhanced immunotherapy utilizing IL-12-DCs presents a hopeful avenue for HCC treatment [66].

It has also been reported that NK cells produce X-C motif chemokine ligand 1 (XCL1), C-C motif ligand 5 (CCL5), and fms-like tyrosine kinase 3 (FLT3) to attract DCs in the HCC TME, thus promoting their antigen presenting function to T cells [39,67].

The enhancement of dendritic cells through vaccines or immunotherapy increases the anti-neoplastic effect due to the establishment of an environment with normal immunitary response [68].

### 3.5. Neutrophils

The recruitment of the neutrophils in the TME takes place through the release of CXCL5 and CXCL6, with the subsequent transformation of the neutrophils in their pro-tumorigenic phenotype (N2 tumor-associated neutrophils—TANs). They have a heavy immunosuppressive action through the liberation of specific cytokines (CCL2, CCL17, IL8) and the expression of PD-L1 [69].

In this way, they are able to enhance both tumor-associated macrophages (TAMs), thus inhibiting the T cells, favoring the resistance to sorafenib and the differentiation of Tregs.

The intense molecular communication between TANs, cancer-associated fibroblasts (CAFs), and neoplastic cells is characterized by the production of cardiotrophin-like cytokine factor 1 by CAFs, which determines the production of CXCL5, TGF-β, and IL-6 and the involvement of the neutrophils, with N2 polarization [70].

In particular, the inhibition of the T-cell population is determined by the hyperactivation of the STAT3 pathway and PD-L1 expression, thus determining the increase in neoplastic stem cells features such as the release of bone morphogenetic protein 2 (BMP2) and TGF-β.

Possible therapeutic pathways in this specific field of the TME are represented by the inhibition of the immunosuppressive action of TANS.

A possible strategy for the treatment of HCC could be the inhibition of TANs’ recruitment through immunotherapies targeting specific neutrophilic antigens [71,72].

### 3.6. Natural Killer Cells

Natural killer (NK) cells are responsible for the triggering of the innate immune response against neoplastic cells and several pathogens [73]. They are characterized by inhibitory receptors (C-type lectin-like receptor NKG2A, inhibitory killer immunoglobulin-like receptors) that could bind HLA-E, as well as classical and non-classical MHC-I complexes. Normal cells are able to inhibit NK cells through these surface receptors (e.g., NKG2A): this propriety is enabled in neoplastic cells, leading to NK cell activation through specific receptors (NKp44, NKG2C/D, NKp30, binding NKP46L, MICA-B, B7H6), which determines a molecular cascade leading to the release of granzyme B and perforin and, finally, to the killing of neoplastic cells [73,74].

The TME is generally characterized by an immunosuppressive milieu that determines NK inactivation: IL-10 signaling leading up with NKG2A expression in NK cells, which is related to the PD-1 expression, culminating with the activation of NK cells and neoplastic expansion. Tumorigenesis is able to block NK cells, but new therapeutic options are being studied to reactivate their functions, such as the stimulation of IL-15, using TGF-β as a target, antibody-dependent cellular cytotoxicity (based on antibodies that are able to bind NK cells and neoplastic cells), chimeric NK cells, and inhibition of CD 96 (NKGA2A) that allows for the blocking of the NK cells with immunosuppressive function [75].

Several studies hinted at the promising results derived from the administration of regorafenib associated with agents that increase NK antineoplastic activity [74,75,76].

### 3.7. An Overview of T Cells in the Hepatocellular Carcinoma Microenvironment

In the dynamic milieu of hepatocellular carcinoma (HCC), the tumor microenvironment (TME) orchestrates intricate interactions with the adaptive immune system, prominently involving CD8+ cytotoxic T cells and CD4+ helper T cells among tumor-infiltrating lymphocytes (TILs). These cells play a crucial role in constraining tumor growth and impeding progression, displaying notable efficacy both within the tumor core and in the peritumoral region [77].

#### 3.7.1. CD8+ T Lymphocytes

At the forefront of cancer immunity, CD8+ T cells, also known as cytotoxic T lymphocytes (CTLs), emerge as pivotal players in identifying and eliminating target cells, encompassing infected and malignant entities. Notably, only a minority of infiltrating lymphoid CD8+ T cells demonstrate tumor recognition capabilities, while the majority remain non-functional [78].

Within the TME, CD8+ T cells interact with diverse immune cell populations, including tumor-associated macrophages (TAMs), myeloid-derived suppressor cells (MDSCs), and regulatory T cells (Tregs), profoundly influencing immune responses and tumor outcomes. However, despite their effector functions, the TME establishes an immunosuppressive milieu characterized by inhibitory cytokine secretion (e.g., transforming growth factor-β (TGF-β), interleukin-10 (IL-10)), expression of immune checkpoint molecules (e.g., PD-L1), and recruitment of immunosuppressive cell populations (e.g., Tregs, MDSCs). These mechanisms ultimately impede the efficacy of CD8+ T cells, fostering immune escape and facilitating tumor progression.

Upon recognition of non-self-antigens presented by dendritic cells (DCs) via the T-cell receptor (TCR), CD8+ T cells undergo activation. This activation involves the binding of CD80−CD86 and CD70 ligands on DCs to CD27 and CD28 receptors on CD8+ T cells, leading to their transformation into cytotoxic effector CD8+ T cells, poised to execute their cytotoxic functions against cancerous targets [79]. Subsequently activated, CD8+ T cells deploy cytotoxic molecules such as perforin, granzymes, and Fas ligand (CD95) to induce target cell apoptosis. Additionally, they secrete cytokines like interferon-gamma (IFN-γ) and tumor necrosis factor-alpha (TNF-α), further contributing to cancer cell demise [80].

Multiple studies emphasize the critical role of CD8+ T cell presence within the TME in orchestrating antitumor immune responses, correlating with favorable outcomes such as increased overall survival and reduced risk of disease recurrence. Chen L. et al. [81] demonstrated enhanced survival rates in HCC patients with elevated levels of CD103-expressing CD8+ T cells. Similarly, Li Z. et al. [82] observed improved prognosis in HCC with high infiltration of CD8+ PD-1+ CD161+ T cells. Additionally, Ye L. et al. [83] illustrated how a subgroup of CXCR5+ CD8+ T cells promotes IgG production in B cells via interleukin-21 (IL-21), eliciting a robust antitumor response.

#### 3.7.2. Employment of CD8+ T Lymphocytes in Immunotherapy

In the realm of cancer immunotherapy, the potent cytotoxic capability of CD8+ T cells is applied to effectively fight against tumor cells. Notably, in individuals with HCC, the expression of immune-checkpoint-associated molecules (PD-1, PD-L1, and CTLA-4) tends to increase due to prolonged chronic inflammation [84,85]. Consequently, this leads to the apoptosis of CD8+ T cells and a reduction in their activity against tumors. Therefore, one of the primary strategies in harnessing CD8+ T lymphocytes involves the inhibition of immune checkpoints.

Immune checkpoints serve as intrinsic mechanisms in the body to prevent the immune system from targeting healthy cells. However, tumors frequently exploit these checkpoints to evade immune surveillance. The utilization of drugs that inhibit these checkpoints, such as anti-PD-1 agents like pembrolizumab and nivolumab, anti-PD-L1 drugs including durvalumab and atezolizumab, and anti-CTLA-4 medications like tremelimumab and ipilimumab, presents promising therapeutic avenues. By doing so, these drugs enable CD8+ T lymphocytes to overcome this blockade, thus enhancing their ability to effectively target tumor cells [86,87].

Currently, among the anti-PD-1 agents, nivolumab received approval from the Food and Drug Administration (FDA) in 2017, based on findings from the open-label, non-comparative phase I/II study CheckMate 040 [88]. Additionally, pembrolizumab was granted FDA approval in 2018 following positive results from the Keynote-224 trial, an open-label phase 2 study that analyzed 104 patients with advanced HCC who had progressed on sorafenib: an ORR of 17% (RECIST v1.1), a PFS of 4.9 months, and a median OS of 12.9 months were reported with pembrolizumab [89]. Both pembrolizumab and nivolumab were approved by the FDA in patients with advanced-stage HCC who have previously undergone sorafenib treatment [90,91].

Presently, there are two additional ongoing phase III trials. The first is the MK-3475-394 study (NCT03062358), which examines the safety and effectiveness of pembrolizumab or placebo administered alongside optimal supportive care as a second-line therapy for advanced HCC patients in Asia. The second trial, MK-3475-937 (NCT03867084), evaluates the efficacy and safety of pembrolizumab compared to placebo as an adjuvant therapy in individuals with HCC who have achieved complete radiological response after local ablation or surgical resection [92,93,94].

Alternative approaches are being explored: for instance, as demonstrated by Huynh JC et al. [95], in a phase I/II study, the combination of BMS-986,205 and nivolumab exhibited a manageable safety profile with sustained benefits as a first-line therapy in a significant subset of advanced HCC patients [95]. In a multicenter phase II trial, simultaneous administration of nivolumab therapy and EBRT revealed encouraging progression-free survival (PFS) outcomes, coupled with acceptable safety profiles, in patients with advanced hepatocellular carcinoma (HCC) and macrovascular invasion [96]. Moreover, in refractory advanced HCC, Yi L et al. illustrated in an open-label, single-arm, pilot study on the efficacy of combination treatment involving oncolytic adenovirus H101 with nivolumab [97].

In their multicenter, open-label, single-group assignment phase I/II study, Wainberg ZA et al. [98] illustrate an acceptable safety profile and promising antitumor activity of durvalumab in patients with advanced HCC who were previously treated with sorafenib. A phase I/IIa trial examining yttrium-90 radioembolization combined with durvalumab for locally advanced unresectable hepatocellular carcinoma has shown encouraging efficacy and safety [99]. Additionally, durvalumab, in conjunction with bevacizumab, is being investigated as an adjuvant therapy in the EMERALD-2 trial (NCT03847428), a randomized, double-blind, placebo-controlled, multicenter, phase III study [92].

Furthermore, the combination of atezolizumab, an anti-PD-L1 agent, with bevacizumab has emerged as the established first-line therapy for advanced HCC, supported by the demonstrated efficacy of this treatment regimen [21,100].

Tremelimumab, a monoclonal antibody targeting CTLA-4, shows significant potential for patients with advanced HCC stemming from HCV-induced liver cirrhosis, as evidenced by the study conducted by Sangro B. et al. [101]. Additionally, Agdashian et al. [102] noted that tremelimumab treatment can stimulate the activation of tumor-specific T cells, decrease T-cell clonality, and enhance the infiltration of CD3+ CD8+ T cells into the tumor, leading to noteworthy clinical and immunological outcomes in patients with HCC.

In a recent publication, Sangro B et al. provided an update from the phase III HIMALAYA study, which examines the combination of tremelimumab and durvalumab in unresectable HCC. This update reinforces the sustained advantages observed in patients with non-resectable hepatocellular carcinoma [103].

The Checkmate 040 randomized clinical trial investigated the effectiveness and safety of ipilimumab in combination with nivolumab, monoclonal antibodies that target CTLA-4 and PD-1 immune checkpoints, respectively, among advanced HCC patients who had previously received sorafenib treatment [104]. Thus far, there have been studies indicating that the combined use of ipilimumab and nivolumab may be efficacious and well-tolerated following prior ICI-based combination therapies. This underscores the need for prospective clinical assessments of this treatment sequence [105,106] (Table 2).

#### 3.7.3. Engineered T-Cell Therapies: CAR-T Therapy

Cancer immunotherapy has witnessed remarkable progress with the advent of engineered T-cell therapies, notably chimeric antigens receptor cells-T (CAR-T) therapy. This innovative approach involves the modification of CD8+ T lymphocytes to express receptors capable of identifying antigens present on tumor cells, thereby augmenting the immune system’s capacity to target and eliminate cancer cells. The glypican 3 gene (GPC3) emerges as a promising target for CAR-T therapy, being a proteoglycan highly expressed in various cancers, including HCC [107,108,109]. Its involvement in HCC development via the Wnt (wingless-type integration site family) signaling pathway renders it an attractive target for immunotherapy [110]. Moreover, the integration of co-stimulatory domains in CAR-T cells, such as CD28 [111,112], can further enhance their cytotoxicity and proliferative potential. In order to optimize therapeutic effectiveness, CAR-T cells expressing cytokines like IL-7 and chemokines like CCL19 have been engineered, significantly enhancing their anti-tumor capabilities [108]. Clinical trials demonstrated the potential of these modified CAR-T cells in eliminating tumors in patients with advanced GPC3+ HCC [108,113]. Additionally, CD133, a glycoprotein associated with cancer stem cells and poor prognosis in HCC patients, represents another potential target for CAR-T therapy. CAR-T cells specifically targeting CD133 have exhibited remarkable efficacy in restraining tumor growth in both preclinical and clinical contexts, underscoring their promise as a treatment avenue for CD133-positive HCC [114].

#### 3.7.4. CD4+ Lymphocytes

The efficient eradication of tumor cells heavily relies on the presence of CD4+ T cells [115]. Their absence can result in a decline in both the quantity and efficacy of tumor-specific T cells [116]. In the context of HCC, the initial stages of the disease witness a notable increase in both circulating and tumor-infiltrating CD8+ and CD4+ T cells, which subsequently decline as the disease advances. Significantly, reduced levels of CD4+ cytotoxic T lymphocytes (CTLs) are associated with shorter survival times and increased mortality rates in HCC patients [116]. This is noteworthy because CD4+ T cells possess inherent cytotoxicity, allowing them to directly target tumor cells even without assistance from CD8+ T cells [117]. Specifically, CD4+ T lymphocytes engage with tumor-associated macrophages (TAMs) in the tumor microenvironment, promoting M1 polarization and initiating an anti-tumor response [118]. The role of CD4+ T cells can also be inferred from translational studies, such as the one conducted by Dhanasekaran R. et al. [119], where an increase in CD4+ lymphocytes in the tumor microenvironment was observed following treatment with antisense oligonucleotides (ASOs) to decrease the expression of the oncogene MYC in murine models of HCC.

Moreover, CD4+ T cells indirectly influence the function of CD8+ T cells within the tumor microenvironment through cytokine secretion and immune regulation. For example, type 1 T helper (Th1) cells secrete cytokines like interleukin-2 (IL-2) and interferon-gamma (IFN-γ) to boost CD8+ T-cell activation and proliferation [120]. Additionally, CD4+ T cells play a crucial role in regulating the activity of other immune cells within the tumor microenvironment, differentiating into subsets such as Th1, type 2 T helper (Th2) cells, type 17 T helper (Th17) cells, and regulatory T cells (Tregs), each with distinct functions. Notably, Th1 cells produce IFN-γ, which enhances the cytotoxic activity of CD8+ T cells and macrophages against cancer cells [121].

##### Regulatory T Cells

Tregs represent a subset of CD4+ T cells whose increased activity in HCC facilitates tumor invasiveness and compromises T-cell immune responses through diverse mechanisms [122]. Notably, CD4+ CD25+ FoxP3+ Treg cells hinder the function of natural killer (NK) cells [123], CD8+ T cells, and Th1 CD4+ T cells by reducing the secretion of cytotoxic molecules like granzymes A and B and perforin, suppressing TNF-α and IFN-γ; expressing CTLA-4; and releasing inhibitory molecules such as interleukin-8 (IL-8), IL-10, interleukin-35 (IL-35), and TGF-β [124]. For all these reasons, elevated levels of circulating CD4+ CD25+ FoxP3+ Tregs are associated with tumor progression and reduced survival rates in patients affected by hepatocellular carcinoma [125], and the FOXP3+ Treg/CD4+ T-cell ratio also serves as a prognostic indicator for overall survival [126]. Another confirmation of the pro-tumorigenic activity of Tregs has been observed by Sawant DV et al., as they have seen that the accumulation of Tregs within the tumor core hinders the infiltration and effector function of CD8+ T cells through upregulation of the “B lymphocyte-induced maturation protein-1” (BLIMP-1) inhibitory receptor axis [127].

For these reasons, the targeting of regulatory T cells emerges as a promising strategy in hepatocellular carcinoma immunotherapy.

As demonstrated by Kalathil SG et al., tivozanib, a tyrosine kinase inhibitor, has exhibited significant efficacy in reducing the presence of Tregs, myeloid-derived suppressor cells (MDSCs), and exhausted T cells [128]. By disrupting the immunosuppressive milieu orchestrated by these regulatory cells, tivozanib holds potential for augmenting anti-tumor immunity and improving treatment outcomes. These findings have also been corroborated by the phase 1b/2 study conducted by Fountzilas C. et al. in patients with advanced HCC, which revealed an early efficacy signal alongside a highly favorable toxicity profile [129]. Currently, a multicentered, phase Ib/II, open-label study of tivozanib with durvalumab in advanced HCC is underway, led by the group of Mahmood S. et al. [130].

Further evidence regarding the therapeutic potential of suppressing Treg activity has been obtained from the study conducted by Liu D et al., which demonstrated in murine models of HCC that sunitinib treatment primes the antitumor immune response. This treatment significantly reduces Treg frequency, diminishes the production of TGF-β and IL-10 by Tregs, and also safeguards tumor-antigen-specific CD8+ T cells from tumor-induced deletion in the context of HCC [131,132].

### 3.8. The Controversial Role of B-Cell Lymphocytes in the HCC Microenvironment

In the context of HCC, significant attention is directed towards the role of B cells within the TME. Presently, conflicting perspectives exist regarding the impact of B cells, leaving uncertainty regarding their influence—whether advantageous or deleterious [133].

On one side, they release cytokines that synergize with cytotoxic T lymphocyte (CTL) activity and serve as robust antigen-presenting cells (APCs) through the expression of CD20, CD80, CD86, and major histocompatibility complex II (MHC-II). These cytokines, including soluble CD19, IL-2, IL-6, IL-12p40, IL-7, IFN-γ, TNF, chemokine (C-C motif) ligand 3 (CCL3), and colony-stimulating factor 2 (CSF2 or GM-CSF), as well as IL-4, promote the differentiation of helper CD4+ T cells into Th1 and Th2 subgroups, modulating immune responses in the HCC TME [133]. As shown by Garnelo M. et al., in murine models of hepatoma cells, the presence of tumor-infiltrating B cells (TIBs) correlated with elevated expression levels of granzyme B and IFN-γ. Depletion of mature B cells in these models resulted in impaired tumor control and reduced local T cell activation, highlighting the significance of the interaction between tumor-infiltrating T cells and B cells in promoting enhanced local immune activation and contributing to a more favorable prognosis for HCC patients [134]. Clinical studies further support these findings, showing that high densities of tumor-infiltrating B cells correlate with improved clinical outcomes in HCC patients. Notably, specific subtypes of B cells such as CD20+ B cells, naïve B cells, and CD27-isotype-switched memory B cells were identified as independent predictors of survival [135]. These findings were further supported by a meta-analysis conducted by Ding et al. [136] on 452 cases, which concluded that patients with elevated densities of CD20+ B cells at the tumor periphery demonstrated increased rates of disease-free and overall survival.

Conversely, evidence suggests that B cells may promote tumorigenesis through the production of cytokines such as IL-10, TGF-β, and IL-35, as well as cytokines that attract myeloid-derived suppressor cells (MDSCs) and facilitate angiogenesis [137]. For instance, it has been demonstrated that chemokine (C-C motif) ligand 20 (CCL20) derived from tumor cells interacts with chemokine receptor 6 (CCR6) highly expressed by CD19+ CD5+ B cells, potentially enhancing angiogenesis and promoting HCC development [138]. Moreover, selective recruitment of regulatory B cells, which express on their surface the chemokine receptor CXCR3 (CXCR3), are implicated in bridging the pro-inflammatory IL-17 response and the polarization of tumorigenic macrophages (M2-type) within the tumor environment, suggesting that inhibiting the migration or function of CXCR3+ B cells could potentially attenuate HCC progression [139,140]. Additionally, CD20+ B cells, as demonstrated by Faggioli et al. [141], may also promote HCC progression through the production of tumor necrosis factor alpha (TNF-α). It is evident from these studies that the tumor-promoting effect of B cells is mainly attributed to the regulation of B cells that secrete immunosuppressive cytokines like IL-10. Moreover, Xiao et al. [142] identified a novel protumorigenic B-cell population (PD-1hi B cells) that, through IL-10-dependent pathways after an interaction with PD-L1, promote T-cell dysfunction and disease progression.

Given these various aspects of the role of B cells in the TME of HCC, it is imperative to clarify the distinct functions of B-cell subpopulations and their interactions with other immune cells to refine targeted immunotherapeutic strategies in HCC.

### 3.9. The Role of Cancer-Associated Fibroblasts (CAFs) in the HCC Microenvironment

Cancer-associated fibroblasts (CAFs) can arise from a spectrum of cell origins, including resident fibroblasts, hepatic stellate cells (HSCs), and bone-marrow-derived mesenchymal stem cells, possessing distinctive functionalities not found in normal fibroblasts [143]. The spatial distribution of CAFs within hepatocellular carcinoma (HCC) tissues exhibits variability, presenting as aggregated, sporadic, or localized configurations along hepatic sinusoids. As demonstrated by Jia C.C. et al., there is a positive correlation between the abundance of CAFs and tumor size in HCC, ultimately contributing to poorer prognostic outcomes for patients with this disease [144].

CAFs play a pivotal role in promoting tumor growth, invasion, metastasis, and angiogenesis through diverse mechanisms. Several studies indicate a two-way communication between tumor cells and stromal cells, underscoring the reciprocal influence between CAFs and tumor progression. Specifically, CAFs secrete growth factors such as transforming growth factor-beta (TGF-β), fibroblast growth factor (FGF), and vascular endothelial growth factor (VEGF), which promote cancer cell proliferation, survival, and angiogenesis [68,145]. Moreover, CAFs contribute to extracellular matrix (ECM) remodeling, fostering a supportive microenvironment for tumor invasion and metastasis. Concurrently, HCC cells possess the capability to activate precursor cells of CAFs, such as hepatic stellate cells (HSCs), through paracrine signaling or exosomes. HCC-derived factors like TGFβ and CXCL6 have been found to activate CAFs, enhancing their secretory function. Mechanistically, molecules such as connective tissue growth factor (CTGF) and platelet-derived growth factor (PDGF) further augment CAF activation, thus promoting tumor progression [143,146].

CAFs wield significant influence over the tumor immune microenvironment (TIME). They play a pivotal role in shaping the polarization of tumor-associated macrophages (TAMs) towards an immunosuppressive M2 phenotype, potentially facilitated by cytokines like IL6, thereby exacerbating HCC progression [147]. Moreover, CAFs modulate tumor-associated neutrophils (TANs), fostering their chemotaxis and upregulating the expression of immunosuppressive molecules such as PD-L1 [68,147].

Furthermore, CAFs exert inhibitory effects on natural killer (NK) cells, dampening their activation and cytotoxicity by secreting immunosuppressive factors like PGE2 [148]. Additionally, CAFs orchestrate the recruitment of dendritic cells (DCs) to the tumor microenvironment, where they undergo education towards a tolerogenic phenotype, fostering T-cell anergy and facilitating regulatory T-cell differentiation [149,150].

Moreover, CAFs play a crucial role in promoting the generation and recruitment of myeloid-derived suppressor cells (MDSCs), potent immunosuppressive entities that impede T-cell function and bolster tumor progression [151].

CAF involvement poses a challenge for HCC treatment. Indeed, as demonstrated by Zhao J. et al. [152], through the secretion of CXCL12, a chemokine that induces the upregulation of FOLR1 in HCC cells, CAFs contribute to resistance to sorafenib therapy. Concurrently, Chen Q et al. have indicated that combination therapy involving PD-1 inhibitors and anti-angiogenic drugs may hold greater promise for patients with elevated CAF abundance compared to single-agent treatment strategies [153].

Several research directions aim to elucidate potential therapies for HCC-targeting CAFs. Yamanaka et al. [154] proposed conophylline as a plausible suppressor of G-protein-coupled receptor 68 (GPR68). Cai J. et al. [155] identified cancer-associated-fibroblast-derived periostin as a major driver of CD51 cleavage, a crucial component in multiple stages of tumor progression, suggesting its utility as a target for therapy in HCC. The research conducted by Lau et al. [156] revealed that hepatocyte growth factor (HGF) derived from cancer-associated fibroblasts (CAFs) regulates liver tumor-initiating cells by activating FRA1 through an Erk1,2-dependent pathway. Consequently, they suggest that targeting the c-Met/FRA1/Hey1 cascade mediated by CAF-derived HGF could serve as a potential therapeutic approach for hepatocellular carcinoma (HCC).

Considering the intricate interplay between CAFs and therapy resistance in HCC, directing interventions towards CAF-mediated mechanisms holds promise for enhancing treatment efficacy. However, further studies are warranted to fully comprehend how to harness CAFs in favor of therapy.

## 4. Conclusions

The continuous evolution of the therapeutic strategy of HCC has led to an in-depth study of the role of the different components of the tumor microenvironment, discovering a complex universe of relationships between the neoplastic cells and the cellular and immunohumoral environment surrounding the tumor. The role of TME is crucial both in the process of tumor initiation and progression, in metastatisation and in resistance to therapies. This intricate intercellular dialogue, in which molecular elements (e.g., cytokines) are also involved, allows us to identify more potential therapeutic paths that could expand HCC management strategies, assisting current treatment methods.

## Figures and Tables

**Figure 1 cancers-16-01837-f001:**
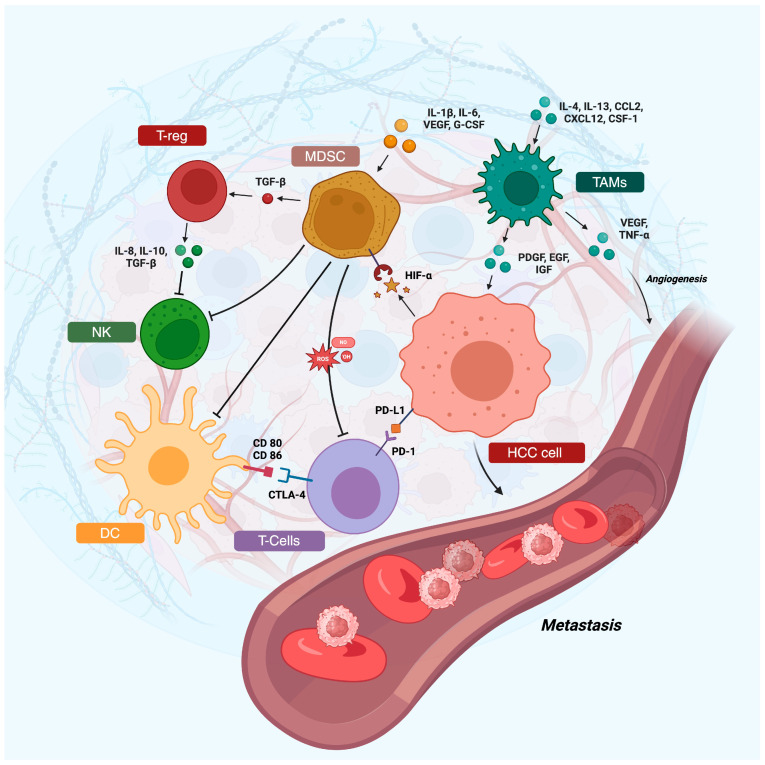
The tumor microenvironment of hepatocellular carcinoma is characterized by the complex interaction between the different cellular populations. Abbreviations: DC: dendritic cells, HCC: hepatocellular carcinoma, NK: natural killer, PD-1: programmed-death-1, PD-L1: programmed-death-ligand 1, T-reg: T-regulatory. Created with BioRender.

**Table 2 cancers-16-01837-t002:** The main studies exploring the employment of CD8+ T lymphocytes in immunotherapy.

Author	Drug	Type of the Study	Aim of the Study	Results
El-Khoueiry et al. [88]	Nivolumab	Open-label non-comparative phase I/II study—CheckMate 040	To assess the safety and efficacy of nivolumab in patients with advanced hepatocellular carcinoma.	- Six-month overall survival was 85% in the HCV-related cohort and 84% in the HBV-related cohort - No treatment-related deaths
Zhu, A.X. et al. [89]	Pembrolizumab	Non-randomised open-label phase 2 trial	To assess the efficacy and safety of pembrolizumab in patients with advanced hepatocellular carcinoma.	- Objective response in 17% of patients with a PFS of 4.9 months and a median OS of 12.9 months - One treatment-related death
Huynh, J.C. et al. [95]	BMS-986,205 + nivolumab	Phase I/II trial	To assess the safety and tolerability of the combination of BMS-986,205 and nivolumab in patients with advanced hepatocellular carcinoma.	- Partial response in 12.5% of patients - Stable disease in 37.5% of patients - Median PFS of 8.5 weeks
Kim, B.H. et al. [96]	Nivolumab + EBRT	Multicenter phase II trial	To assess the efficacy and safety of concurrent nivolumab and EBRT in HCC with macrovascular invasion.	- Median PFS of 5.6 months - Median overall survival of 15.2 months - Median time-to-progression of 5.6 months - Median duration of response of 9.9 months - No treatment-related deaths
Yi, L. et al. [97]	Adenovirus H101 + nivolumab.	Open-label, single-arm, pilot study	To assess the efficacy of combination of adenovirus H101 and nivolumab in refractory advanced hepatocellular carcinoma.	- ORR of 11.1% - Disease control rate of 38.9% - Six-month survival rate of 88.9% - Median progression-free survival of 2.69 months - Overall survival of 15 months
Lee, Y.B. et al. [99]	Yttrium-90 radioembolization combined with durvalumab	Phase I/IIa pilot trial	To assess the efficacy and safety of combination of radioembolization with yttrium-90 microspheres (Y90-radioembolization) and durvalumab in patients with locally advanced unresectable HCC.	- Median time-to-progression of 15.2 months - Median progression-free-survival of 6.9 months - 29.2% of complete response - 54.2% of partial response - ORR of 83.3% - No treatment-related death
Galle, P.R. et al. [100]	Atezolizumab + bvacizumab	Open-label, randomized, phase 3 trial	To assess the patient-reported outcomes with atezolizumab and bevacizumab in advanced hepatocellular carcinoma.	- Reduction of the risk of deterioration in all EORTC QLQ-C30 generic cancer symptom scales
Sangro, B. et al. [101]	Tremelimumab	Pilot clinical trial	To assess the antitumor and antiviral effect of tremelimumab in HCC and HCV infection and to assess the safety of it in cirrhotic patients.	- Partial response rate of 17.6% - Disease control rate of 76.4% - Time to progression of 6.48 months - Antiviral effect
Sangro, B. et al. [103]	Tremelimumab + darvalumab	Update from the phase III HIMALAYA study	To assess the efficacy of tremelimumab plus durvalumab in unresectable hepatocellular carcinoma.	- 30.7% of 36 months OS - 25.2% of 48 months OS
Yau, T. et al. [104]	Ipilimumab + nivolumab	Randomized clinical trial	To assess the efficacy and safety of nivolumab and ipilimumab in patients with advanced hepatocellular carcinoma.	- Response rates of 32%/27%/29%, respectively - One treatment-related death in one group

Abbreviations: EBRT: external-beam radiation therapy, HBV: hepatitis B virus, HCC: hepatocellular carcinoma, HCV: hepatitis C virus, ORR: overall response rate, OS: overall survival, PFS: progression free survival.

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
