# Peer review of "Hepatocellular Carcinoma and the Multifaceted Relationship with Its Microenvironment: Attacking the Hepatocellular Carcinoma Defensive Fortress"

_cancers, 2024, doi:10.3390/cancers16101837_

Round 1

Reviewer 1 Report

Comments and Suggestions for Authors

The manuscript entitled " HCC and the multifaceted relationship with its microenvironment: attacking HCC defensive fortress. " was reviewed.

This paper is well organized. What is the authors' main focus in HCC? The authors' approach should be included.

Comments on the Quality of English Language

none

Author Response

ANSWER TO REVIEWER 1: Dear Doctors, we would like to thank you for your observations; we would like to underline that our paper is  focalized on the microenvironment of hepatocellular carcinoma and on the possible implications that this really complex system could have in the context of tumor treatment (page 2, line 74-78).

Reviewer 2 Report

Comments and Suggestions for Authors

The paper is well written.

Treatment modalities such as surgical resection, radiofrequency ablation, and 66 transarterial chemoembolization are effective for liver-located HCC.(line 67-68). 

The statement could be more comprehensive such as transplant and radiotherpay.

Author Response

ANSWER TO REVIEWER 2: We are grateful for your kind attention and suggestions. As per your advice, we added “transplantation and radiotherapy“ to HCC treatments (page 2, line 66-67).

Reviewer 3 Report

Comments and Suggestions for Authors

This is a well-written review that described in details the cellular interactions between HCC and its microenvironment.

There is an issue with tables 1 and 2.

Author Response

ANSWER TO REVIEWER 3: Dear Doctors, we corrected the issue about the second table and we added it to our paper (pages 15-16).